# Fibromyalgia: Recent Advances in Diagnosis, Classification, Pharmacotherapy and Alternative Remedies

**DOI:** 10.3390/ijms21217877

**Published:** 2020-10-23

**Authors:** Massimo E. Maffei

**Affiliations:** Department of Life Sciences and Systems Biology, University of Turin, 10135 Turin, Italy; massimo.maffeil@unito.it; Tel.: +39-011-670-5967

**Keywords:** fibromyalgia, diagnosis, pharmacotherapy, alternative therapies, plant extracts, natural products

## Abstract

Fibromyalgia (FM) is a syndrome that does not present a well-defined underlying organic disease. FM is a condition which has been associated with diseases such as infections, diabetes, psychiatric or neurological disorders, rheumatic pathologies, and is a disorder that rather than diagnosis of exclusion requires positive diagnosis. A multidimensional approach is required for the management of FM, including pain management, pharmacological therapies, behavioral therapy, patient education, and exercise. The purpose of this review is to summarize the recent advances in classification criteria and diagnostic criteria for FM as well as to explore pharmacotherapy and the use of alternative therapies including the use of plant bioactive molecules.

## 1. Introduction

Fibromyalgia (FM) (earlier considered to be fibrositis, to stress the role of peripheral inflammation in the pathogenesis) is a syndrome that does not present a well-defined underlying organic disease. The primary driver of FM is sensitization, which includes central sensitivity syndromes generally referred to joint stiffness, chronic pain at multiple tender points, and systemic symptoms including cognitive dysfunction, sleep disturbances, anxiety, fatigue, and depressive episodes [1,2]. FM is a heterogeneous condition that is often associated to specific diseases such as infections, psychiatric or neurological disorders, diabetes and rheumatic pathologies. FM is more frequent in females, where it causes musculoskeletal pain [3] and affects significantly the quality of life, often requiring an unexpected healthcare effort and consistent social costs [4,5]. Usually, a patient-tailored approach requires a pharmacological treatment by considering the risk-benefit ratio of any medication. Being the third most common diagnosis in rheumatology clinics, FM prevalence within the general population appears to range from 1.3–8% [2]. To date there are no specific tests specific for FM. FM is currently recognized by the widespread pain index (which divides the body into 19 regions and scores how many regions are reported as painful) and a symptom severity score (SSS) that assesses cognitive symptoms, unrefreshing sleep and severity of fatigue [6]. It is not clear what causes FM and diagnosing assist the patients to face polysymptomatic distress, thereby reducing doubt and fear which are main psychological factors contributing to this central amplification mechanism [7]. In this review, an update on diagnosis and therapy of FM is provided along the discussion on the possibility of using pharmacological drugs, bioactive natural substances and alternative therapies to alleviate the symptomatology in combination or as alternative remedies to drugs.

## 2. Diagnosis

To date there is still a considerable controversy on the assessment and diagnosis of FM. Despite advances in the understanding of the pathologic process, FM remains undiagnosed in as many as 75% of people with the condition [8].

The first attempt for the FM classification criteria is dated 1990 and is based on studies performed in 16 centers in the U.S.A. and Canada in clinical and academic settings, gathering the both doubters and proponents [9]. Since then, several alternative methods of diagnosis have been proposed. In general, most of the researchers agree on the need to assess multiple domains in FM including pain, sleep, mood, functional status, fatigue, problems with concentration/memory (i.e. dyscognition) and tenderness/stiffness [5]. Four core areas were initially assessed: (1) pain intensity, (2) physical functioning, (3) emotional functioning, and (4) overall improvement/well-being [10]. About 70–80% of patients with FM also report having sleep disturbances and fatigue. Depressive symptoms, anxiety and mood states have also been included in FM diagnosis. An impairment in multiple areas of function, especially physical function is often reported by patients with FM [11] with a markedly impaired function and quality of life [8]. Since the late 1990′s, a top priority was the development of new disease-specific measures for each of the relevant domains in FM. Also, much attention was paid to studies supporting the valid use of existing instruments specifically in the context of FM [5]. 

Later on, in 2010, the tender point count was abandoned and the American College of Rheumatology (ACR) suggested preliminary diagnostic criteria which were considering the number of painful body regions evaluating the presence and severity of fatigue, cognitive difficulty, unrefreshed sleep and the extent of somatic symptoms. The diagnostic criteria are not based on laboratory or radiologic testing to diagnose FM and rely on a 0–12 Symptom Severity Scale (SSS) which is used to quantify FM-type symptom severity [12]. Furthermore, the SSS was proposed to be combined with the Widespread Pain Index (WPI) into a 0–31 Fibromyalgianess Scale (FS) [13]. With a specificity of 96.6% and sensitivity of 91.8%, a score ≥ 13 for FS was able to correctly classify 93% of patients identified as having FM based on the 1990 criteria [14]. ACR 2010 criteria were also found to be more sensitive than the ACR 1990 criteria, allowing underdiagnosed FM patients to be correctly identified and giving a treatment opportunity to those who had previously been untreated [15]. It is still unclear whether the diagnosis of FM has the same meaning with respect to severity in primary FM (PFM, a dominant disorder that occurs in the absence of another clinically important and dominant pain disorder) and secondary FM (SFM, which occurs in the presence of another clinically important and dominant medical disorder) [16]. Figure 1 shows the ACR 1990 criteria for the classification of fibromyalgia, whereas Figure 2 shows a graphical representation of the Symptom Severity Scale (SSS) plus the Extent of Somatic Symptoms (ESS).

Table 1 shows a holist approach based on the assumption that a multitude of potential diagnoses is fundamental in order to avoid an FM misdiagnosis [17].

In 2013, alternative diagnostic criteria have been developed by some clinicians in the USA including more pain locations and a large range of symptoms than ACR 2010. A self-reported survey was composed of the 28-area pain location inventory and the 10 symptom items from the Symptom Impact Questionnaire (SIQ) [18]. However, when compared to the early 2010 criteria, these alternative criteria did not significantly contribute in differentiating common chronic pain disorders from FM [1].

In 2015, the view of diagnostic criteria was altered by ACR by providing approval only for classification criteria and no longer considering endorsement of diagnostic criteria, stressing that diagnostic criteria are different from classification criteria and are beyond the remit of the ACR [19]. However, the suggestion that diagnostic and classification criteria represent 2 ends of a continuum implies that the continuum represents the accuracy of the criteria [20]. Classification criteria and diagnostic criteria could intersect; however, according to some authors the terms “diagnosis” and “classification criteria” should be considered as qualitatively distinct concepts. The proposed concept of “diagnostic criteria” [19] is challenging and may be hardly realizable, while diagnostic guidelines based on proper modelling techniques may be helpful for clinicians in particular settings [20].

In 2016, based on a generalized pain criterion and clinic usage data, a new revision of the 2010/2011 FM criteria was developed including the following criteria: 1) generalized pain, defined as pain present in at least 4 of 5 regions; 2) symptoms present at a similar level for at least three months; 3) a WPI ≥ 7 and SSS ≥ 5 or WPI of 4–6 and SSS ≥ 9; 4) a diagnosis of FM is valid irrespective of other diagnoses. Another important point is that the presence of other clinically important illnesses does not exclude a diagnosis of FM [21].

In 2018, considering important but less visible factors that have a profound influence on under- or over-diagnosis of FM provided a new gate to a holistic and real understanding of FM diagnosis, beyond existing arbitrary and constructional scores [22].

In 2019, in cooperation with the WHO, an IASP Working Group has developed a classification system included in the International Classification of Diseases (ICD-11) where FM has been classified as chronic primary pain, to distinguish it from pain which is secondary to an underlying disease [23].

More recently, a study of about 500 patients under diagnosis of FM, revealed that 24.3% satisfied the FM criteria, while 20.9% received a clinician International Classification of Diseases (ICD) diagnosis of FM, with a 79.2% agreement between clinicians and criteria. The conclusions of this study pointed out a disagreement between ICD clinical diagnosis and criteria-based diagnosis of FM, calling into question meaning of a FM diagnosis, the validity of physician diagnosis and clinician bias [24].

FM is a disorder that cannot be based on diagnosis of exclusion, rather needing positive diagnosis [6], through a multidimensional FM diagnostic approach making diagnosis encompassing psychosocial stressors, subjective belief, psychological factors and somatic complaints [25]. The advent of the PSD scale identified a number of problems in FM research [16].

Recently, immunophenotyping analysis performed on blood samples of FM patients revealed a role of the Mu opioid receptor on B lymphocytes as a specific biomarker for FM [26]. Moreover, a rapid biomarker-based method for diagnosing FM has been developed by using vibrational spectroscopy to differentiate patients with FM from those with other pain-related diseases. Unique IR and Raman spectral signatures were correlated with FM pain severity measured with FM impact questionnaire revised version (FIQR) [27]. Overall, these findings provide reliable diagnostic tests for differentiating FM from other disorders, for establishing serologic biomarkers of FM-associated pain and were useful for the contribution of the legitimacy of FM as a truly painful disease.

In summarizing aspects of FM learned through applications of criteria to patients and trials, Wolfe [28] identified 7 main concepts: 1) there is no way of objectively testing FM which also has no binding definition; 2) prevalence and acceptance of FM depend on factors largely external to the patient; 3) FM is a continuum and not a categorical disorder; 4) every feeling, symptom, physical finding, neuroscience measure, cost and outcome tells one very little about the disorder and its mechanisms when fibromyalgia to “normal subjects” is compared; 5) the range and content of symptoms might indicate that FM may not truly be a syndrome; 6) “pain and distress” type of FM subject identified in the general population [29] might be considered as part of the FM definition and; 7) caution is needed when accepting the current reductive neurobiological causal explanations as sufficient, since FM is a socially constructed and arbitrarily defined and diagnosed dimensional disorder.

## 3. Therapy

### 3.1. Pharmacotherapy of FM

Clinical trials have failed to conclusively provide overall benefits of specific therapies to treat FM; therefore, current pharmacological treatments for patients suffering from FM are mainly directed to palliate some symptoms, with relevant clinical benefits experienced only by a minority of individuals from any one intervention. In those treated with pharmacotherapy, a 50%reduction in pain intensity is generally achieved only by 10% to 25% [30] However, some treatments seem to significantly improve the quality of life of certain FM patients [31]. Only a few drugs have been approved for use in the treatment of FM by the US FDA, whereas no drug has been approved for this indication by the European Medicines Agency. Thus patients with FM frequently need to be treated on an off-label basis [32].

Currently, only 25% to 40% pain reduction is granted by drugs and meaningful relief occurs in only 40% to 60%, in part due to dose-limiting adverse effects and incomplete drug efficacy [33]. These limitations in clinical practice have led some to hypothesize that a combination of different analgesic drugs acting through different mechanisms may provide superior outcomes compared to monotherapy [34]. Moreover, drugs should be started at low doses and cautiously increased because some patients, either do not tolerate or benefit from drug therapy. Because sleep disturbance, pain and psychological distress are the most amenable to drug therapy, drugs should be chosen to manage the individual’s predominant symptoms [35]. Currently, several drugs are frequently used alone or in combination to manage FM symptoms; however, the US FDA indicated for FM only three: two selective serotonin and norepinephrine reuptake inhibitors (SNRIs), duloxetine and milnacipran, and an anticonvulsant, pregabalin [36]. In the next sections, the use of selected drugs aimed to alleviate FM will be described.

#### 3.1.1. Cannabinoids in FM Therapy

The cannabinoid system is ubiquitous in the animal kingdom and plays multiple functions with stabilizing effects for the organism, including modulation of pain and stress, and the management of FM may have therapeutic potential by manipulating this system. The cannabinoid system contributes in maintaining equilibrium and stabilizing effects on FM [37]. Moreover, the endocannabinoid neuromodulatory system is involved in multiple physiological functions, such as inflammation and immune recognition, endocrine function, cognition and memory, nausea, antinociception and vomiting, [38]. Deficiency in the endocannabinoid system has been correlated to FM [39], but without clear clinical evidence in support of this assumption [40].

The endocannabinoid system consists of two cannabinoid receptors, the CB1 and CB2 receptors [41]. In acute and chronic pain models, analgesic effects are associated to CB1 agonists that act at many sites along pain transmission pathways, including activation of spinal, supraspinal and peripheral CB1 receptors, each independently decreasing nociception [42]. Delta 9-tetrahydrocannabinol (Δ9-THC or Dronabinol, **1**) is the main active constituent of *Cannabis sativa* var *indica*, with psychoactive and pain-relieving properties. The non-selective binding to G-protein-coupled CB receptors is responsible for the pharmacological effects induced by Δ9-THC. Cannabidiol (CBD, **2**), a non-psychotropic constituent of cannabis, is a high potency antagonist of CB receptor agonists and an inverse agonist at the CB2 receptor [43]. CBD displays CB2 receptor inverse agonism, an action that appears to be responsible for its antagonism of CP55940 at the human CB2 receptor [44]. This CB2 receptor inverse agonist ability of CBD may contribute to its documented anti-inflammatory properties [44]. The main endocannabinoids are anandamide (*N*-arachidonoylethanolamine, AEA, **3**) and 2-arachidonoylglycerol (2-AG, **4**), AG), the activity of which is modulated by the hydrolyzing fatty acid palmitoylethanolamide (PEA, **5**) and the endocannabinoid precursor arachidonic acid (AA, **6**) [45]. AEA and 2-AG are functionally related to Δ9-THC [46]. It was found that stress induces a rapid anandamide release in several CNS regions resulting in stress-induced analgesia via CB1 receptors [47]. FM patients had significantly higher anandamide plasma levels [39,46]; however, it has been suggested that the origin of FM and chronic pain depend on a deficiency in the endocannabinoid signaling [45].

Monotherapies of FM based on Δ9-THC are based on the assumption that this compound acts as an analgesic drug; however, although a sub-population of FM patients reported significant benefits from the use of Δ9-THC, this statement cannot be made [48]. When the quality of life of FM patients who consumed cannabis was compared with FM subjects who were not cannabis users, a significant improvement of symptoms of FM in patients using cannabis was observed, although there was a variability of patterns [49].

The synthetic cannabinoid nabilone (**7**) showed of a superiority over placebo to reduce FM symptoms, with significant reductions in Visual Analog Scale (VAS) for pain, FM Impact Questionnaire (FIQ), and anxiety [42], indicating the efficacy of treating people with FM with nabilone. Nabilone was also effective in improving sleep [50]; however, participants taking nabilone experienced more adverse events (such as dizziness/drowsiness, dry mouth and vertigo) than did participants taking placebo or amitriptyline (see below).

The self-medication practice of herbal cannabis was associated with negative psychosocial parameters. Therefore, caution should be exercised in recommending the use of cannabinoids pending clarification of general health and psychosocial problems [51,52]. Figure 3 illustrates the chemical formulas of some cannabinoids and endocannabinoids.

#### 3.1.2. Opioids in FM Therapy

One of the major natural sources of opioids is the medicinal plant *Papaver somniferum*. Although clinical evidence demonstrating the efficacy or effectiveness of opioids analgesics is scanty, these molecules are widely used for the treatment of FM [53]. However, the long-term use of opioids in FM has been discouraged by several medical guidelines [54]. The use of opioids is documented in studies demonstrating increased endogenous opioid levels in the cerebrospinal fluid of patients with FM vs. controls [55]. These results prompted the interesting hypothesis that a more activated opioid system can be detected in individuals with FM, reflecting reduced receptor availability and increased release of endogenous opioids [54].

There is evidence from both single center, prospective, longitudinal and multicenter and observational clinical studies of negative effects of the use of opioids in FM on patient outcomes compared with other therapies [56,57]. Moreover, opioid user groups showed less improvement in the SFM-36 subscale scores of general health perception and in the FIQ subscale scores of job ability, fatigue and physical impairment [58]. Furthermore, altered endogenous opioid analgesic activity in FM has been demonstrated and suggested as a possible reason for why exogenous opiates appear to have reduced efficacy [59]. Despite these facts, opioids have been prescribed for 10% to 60% of patients with FM as reported in large database sets [54].

When considered, the preference of patients appears towards opioids. In a survey, 75% of patients considered hydrocodone (**8**) plus acetaminophen to be helpful, and 67% considered oxycodone (**9**) plus acetaminophen to be helpful [60]. FM has been associated with preoperative opioid use, including hydrocodone [61], whereas there is limited information from randomized controlled trials on the benefits or harms of oxycodone when used to treat pain in FM [62].

A pilot study showed that naltrexone (**10**) reduced self-reported symptoms of FM (primarily daily pain and fatigue) [63] and further studies showed that low-dose naltrexone had a specific and clinically beneficial impact on FM. This opioid, which is widely available and inexpensive, was found to be safe and well-tolerated. Blocking peripheral opioid receptors with naloxone (**11**) was observed to prevent acute and chronic training-induced analgesia in a rat model of FM [64]; however, there were no significant effects of naloxone nor nocebo on pressure pain threshold, deep tissue pain, temporal summation or conditioned pain modulation in chronic fatigue syndrome/FM patients [65].

A synthetic opioid receptor agonist that shows serotonin-norepinephrine reuptake inhibitor properties is tramadol (**12**); this compound is often prescribed for painful conditions [66]. Tramadol has been studied in humans who suffer from FM [56], suggesting that tramadol may be effective in treating FM [67]. The use of tramadol provides change in pain assessed by visual analogue scale and FM impact questionnaire; however, the reported side effects include dizziness, headache, constipation, addiction, withdrawal, nausea, serotonin syndrome, somnolence, pruritus seizures, drug–drug interactions with antimigraine and antidepressants medications [66]. Therefore, it is recommended that tramadol application should be considered in refractory and more treatment-resistant cases of FM.

Another weak opioid is codeine (**13**). In a comparative study, there was a significantly higher proportion of patients in the codeine-acetaminophen group reporting somnolence or constipation and a larger proportion of patients in the tramadol-acetaminophen group reporting headache. The overall results suggested that tramadol-acetaminophen tablets (37.5 mg/325 mg) were as effective as codeine-acetaminophen capsules (30 mg/300 mg) in the treatment of chronic pain [68].

Fentanyl (**14**) works primarily by activating μ-opioid receptors and was found to be around 100 times stronger than morphine (**15**), although its effects are more localized. Fentanyl injections reduced second pain from repeated heat taps in FM patients. Similar to reports of effects of morphine on first and second pain, fentanyl had larger inhibitory effects on slow temporal summation of second pain than on first pain from a nociceptor stimulation [69]. Since fentanyl can inhibit windup of second pain in FM patients, it can prevent the occurrence of intense summated second pain and thereby reduce its intensity by a greater extent than first or second pains evoked by single stimuli. Among the 70,237 drug-related deaths estimated in 2017 in the US, the sharpest increase occurred among those related to fentanyl analogs with almost 29,000 overdose deaths which represents more than 45% increase from 2016 to 2017 [70]. Because the numbers of overdoses and deaths due to fentanyl will continue to increase in the coming years, studies are needed to elucidate the physiological mechanisms underlying fentanyl overdose in order to develop effective treatments aimed to reduce the risk of death [71].

Glial cell activation is one of the several other possible pathophysiologic mechanisms underlying the development of FM by contributing to central nervous system sensitization to nociceptive stimuli [72]. Pentoxifylline (**16**), a xanthine derivative used as a drug to treat muscle pain in people with peripheral artery disease, is a nonspecific cytokine inhibitor that has been shown to attenuate glial cell activation and to inhibit the synthesis of TNFα, IL-1β, and IL-6 [73]. In theory, attenuating glial cell activation via the administration of pentoxifylline to individuals suffering from FM might be efficient in ameliorating their symptoms without being a globalist therapeutic approach targeting all possible pathophysiologic mechanisms of development of the syndrome [74]. With regards FM pathophysiology, serum brain-derived neurotrophic factors (BDNF) were found at higher levels in FM patients while BDNF methylation in exon 9 accounted for the regulation of protein expression. These data suggest that altered BDNF levels might represent a key mechanism explaining FM pathophysiology [75].

Opioid users were also observed to experience a decreased pain and symptom severity when caffeine (**17**) was consumed, but this was not observed in opioid nonusers, indicating caffeine may act as an opioid adjuvant in FM-like chronic pain patients. Therefore the consumption of caffeine along with the use of opioid analgesics could represent an alternative therapy with respect to opioids or caffeine alone [76]. Figure 4 shows the chemical formulae of some opioids used in FM therapy.

#### 3.1.3. Gabapentinoids in FM Therapy

Gabapentinoid drugs are US Food and Drug Administration (FDA) (but not in Europe) anticonvulsants approved for treatment of pain syndromes, including FM. However, FDA approved pregabalin (**18**) but not gabapentin (**19**) for FM treatment; nevertheless, gabapentin is often prescribed off-label for FM, presumably because it is substantially less expensive [77]. Pregabalin is a gamma-aminobutyric acid (GABA) analog and is a ligand for the α2δ subunit of the calcium channel being able of reducing the ability of docked vesicles to fuse and release neurotransmitters [78]. Pregabalin shows effects on cortical neural networks, particularly when basal neurons are under hyperexcitability. The pain measures and pregabalin impact on the cortical excitability was observed only in FM patients [79]. Pregabalin was also found to increase norepinephrine levels in reserpine-induced myalgia rats [80]. Because of its tolerability when used in combination with antidepressants, pregabalin use showed a very good benefit to risk ratio [81]. The starting approved dosage for pregabalin is at 150 mg daily [82]; however, the drug shows a higher effectiveness when used at a dose of 300 or 600 mg/day. Lower pregabalin doses than those of clinical trials are used in clinical practice because higher doses are more likely to be intolerable [83]. A recent systematic review shows that a minority of people with moderate to severe pain due to FM treated with a daily dose of 300 to 600 mg of pregabalin had a reduction of pain intensity over a follow-up period of 12 to 26 weeks, with tolerable adverse effects [84]. Thus, pregabalin is one of cardinal drugs used in the treatment of FM, and its clinical utility has been comprehensively demonstrated [85,86]. Nevertheless, there is still insufficient evidence to support or refute that gabapentin may reduce pain in FM [87]. Figure 5 depicts the chemical formulae of some gabapentinoids.

#### 3.1.4. Serotonin–Norepinephrine Reuptake Inhibitors in FM Therapy

There is a wide use of serotonin and noradrenaline reuptake inhibitors (SNRIs). There is no unbiased evidence that serotonin selective reuptake inhibitors (SSRIs) are superior to placebo in treating depression in people with FM and for treating the key symptoms of FM, namely sleep problems, fatigue and pain. However, it should be considered that young adults aged 18 to 24, with major depressive disorder, showed an increased suicidal tendency when treated with SSRIs [88]. A recent Cochrane review evaluated the use of SNRIs including eighteen studies with a total of 7,903 adults diagnosed with FM, by using desvenlafaxine (**20**) and venlafaxine (**21**) in addition to duloxetine (**22**) and milnacipran (**23**), by considering various outcomes for SNRIs including health related quality of life, fatigue, sleep problems, pain and patient general impression, as well as safety and tolerability [89]. Fifty two percent of those receiving duloxetine and milnacipran had a clinically relevant benefit over placebo compared to 29% of those on placebo, with much or very much improvements in the intervention. On the other hand, reduction of pain intensity was not significantly different from placebo when desvenlafaxine was used. However, pain relief and reduction of fatigue was not clinically relevant for duloxetine and milnacipran in 50% or greater and did not improve the quality of life [90]. Same negative outcomes were found for reducing problems in sleep and the potential general benefits of duloxetine and milnacipran were outweighed by their potential harms.

The efficacy of venlafaxine in the treatment of FM was studied to a lesser extent. The lack of consistency in venlafaxine dosing, placebo control and blinding make difficult to understand whether the molecule is effective in treating FM. Nevertheless, tolerability and the lower cost of venlafaxine increases its potential use for the treatment of FM, by rendering the molecule a more affordable option compared to the other, more expensive SNRIs [91].

Mirtazapine (**24**) promotes the release of noradrenaline and serotonin by blocking α_2_-adrenergic autoreceptors and α_2_-adrenergic heteroreceptors, respectively. Mirtazapine, by acting through 5-HT_1A_ receptors and by blocking postsynaptic 5-HT_2A_, 5-HT_2C_, and 5-HT_3_ receptors is able to enhance serotonin neurotransmission [92]. For these properties, mirtazapine is classified as a noradrenergic and specific serotonergic antidepressant [93]. Mirtazapine appears to be a promising therapy to improve sleep, pain, and quality of life in patients with FM [94]. In Japanese patients with FM, mirtazapine caused a significantly greater reduction in the mean numerical rating scale pain score and remained significantly greater from week 6 onward, compared with placebo. However, Adverse mirtazapine caused adverse events including weight gain, somnolence and increased appetite when compared to placebo [92].

Among antidepressants, the tricyclic antidepressant (TCAs) amitriptyline (**25**) was studied more than other antidepressants. It is frequently used to assess comparative efficacy [95] and for many years amitriptyline has been a first-line treatment for FM. Although there is no supportive unbiased evidence for a beneficial effect, the drug was successful for the treatment in many patients with FM. However, amitriptyline achieve satisfactory pain relief only by a minority of FM patients and is unlikely that any large randomized trials of amitriptyline will be conducted in FM to establish efficacy statistically, or measure the size of the effect [96]. Figure 6 depicts the chemical formulae of some SNRIs and TCA.

### 3.2. Alternative Therapies for FM

A survey of the European guidelines shows that most of the pharmacological therapies are relatively modest providing only weak recommendations for FM [97]. A multidimensional approach is therefore required for the management of FM, including pharmacological therapies along with behavioral therapy, exercise, patient education and pain management. A multidisciplinary approach combines pharmacotherapy with physical or cognitive interventions and natural remedies. Very often, patients seek help in alternative therapies due to the limited efficacy of the therapeutic options. The following sections discuss some of the most used alternative therapies to treat FM.

#### 3.2.1. Acupunture

Acupuncture shows low to moderate-level in improving pain and stiffness in people with FM. In some cases, acupuncture does not differ from sham acupuncture in improving sleep or global well-being or reducing pain or fatigue. The mechanisms of acupuncture action in FM treatment appears to be correlated to changes in serum serotonin levels [98]. Electro-acupuncture (EA) was more effective than manual acupuncture (MA) for improving sleep, global well-being and fatigue and in the reduction of pain and stiffness. Although effective, the effect of acupuncture is not maintained at six months follow-up [99]. Moreover, there is a lack of evidence that real acupuncture significantly differs from sham acupuncture with respect to improving the quality of life, both in the short and long term. However, acupuncture therapy is a safe treatment for patients with FM [100,101].

#### 3.2.2. Electric Stimulation

As we discussed, FM, aside pain, is characterized by anxiety, depression and sleep disturbances, and by a complex cognitive dysfunctioning status known as “fibrofog” which is characterized by disturbance in working memory, attention and executive functions globally often referred by the patients as a sense of slowing down, clumsiness and confusion that have a profound impact on the ability to perform and effectively plan daily activities [102,103]. Besides stimulation with acupuncture, the effective modulation of brain areas has been obtained through non-invasive brain stimulation by magnetic or electric currents applied to the scalp like transcranial magnetic and electrical stimulation. In many cases, to relieve pain and improve general FM-related function, the use of anodal transcranial direct current stimulation over the primary motor cortex was found to be significantly more effective than sham transcranial direct current stimulation [104]. If we consider that pharmacological and non-pharmacological treatments are often ineffective or transitory in their effect on FM, therapeutic electrical stimulation appears to have a potential role [105]. Cognitive functions such as memory have been enhanced in FM patients by anodal transcranial direct current stimulation over the dorsolateral prefrontal cortex and has clinical relevance for top-down treatment approaches in FM [106]. In FM patients, modulation of hemodynamic responses by transcutaneous electrical nerve stimulation during delivery of nociceptive stimulation was also investigated and shown to be an effective factor in FM treatment, although the underlying mechanism for these findings still needs to be clarified [107]. It has been recently demonstrated that both transcutaneous electric nerve stimulation and acupuncture applications seem to be beneficial in FM patients [108].

In a recent Positron Emission Tomography H_2_^15^O activation study it was shown that occipital nerve field stimulation acts through activation of the descending pain inhibitory pathway and the lateral pain pathway in FM, while electroencephalogram shows activation of those cortical areas that could be responsible for descending inhibition system recruitment [109].

Microcirculation is of great concern in patients with FM. Recently low-energy pulsed electromagnetic field therapy was found to increase a promising therapy to increase microcirlulation [110]; however, neither pain and stiffness were reduced nor functioning was improved by this therapy in women with FM [111].

The European Academy of Neurology, based on the method of GRADE (Grading of Recommendations, Assessment, Development, and Evaluation) judged anodal transcranial direct currents stimulation of motor cortex as still inconclusive for treatment of FM [112]. Therefore, further studies are needed to determine optimal treatment protocols and to elucidate the mechanisms involved [113].

#### 3.2.3. Vibroacoustic and Rhythmic Sensory Stimulation

Stimulation with sensory events such as pulsed or continuous auditory, vibrotactile and visual flickering stimuli are referred as rhythmic sensory stimulation [114].

Clinical studies have reported the application of vibroacoustic stimulation in the treatment of FM. In a clinal study, one group of patients with FM listened to a sequence of Bach’s compositions, another was subjected to vibratory stimuli on a combination of acupuncture points on the skin and a third group received no stimulation. The results showed that a greater effect on FM symptoms was achieved by the combined use of music and vibration [115]. However, in another study, neither music nor musically fluctuating vibration had a significant effect on tender point pain in FM patients when compared to placebo treatment [116]. Because thalamocortical dysrhythmia is implicated in FM and that low-frequency sound stimulation can play a regulatory function by driving neural rhythmic oscillatory activity, volunteers with FM were subjected to 23 min of low-frequency sound stimulation at 40 Hz, delivered using transducers in a supine position. Although no adverse effects in patients receiving the treatment, no statistically and clinically relevant improvement were observed [117]. On the other hand, gamma-frequency rhythmic vibroacoustic stimulation was found to decrease FM symptoms (depression, sleep quality and pain interference) and ease associated comorbidities (depression and sleep disturbances), opening new avenues for further investigation of the effects of rhythmic sensory stimulation on chronic pain conditions [118].

#### 3.2.4. Thermal Therapies

Thermal therapies have been used to treat FM. Two main therapies are currently used: body warming and cryotherapy.

Because FM is strongly linked to rheumatic aches, the application of heat by spa therapy (balneotherapy) appears as a natural choice for the treatment of FM [119]. Spa therapy is a popular treatment for FM in many European countries, as well as in Japan and Israel. A randomized prospective study of a 10-day treatment was done on 48 FM patients improving their quality of life [120] and showed that treatment of FM at the Dead Sea was both effective and safe [121]. FM patients who were poorly responding to pharmacological therapies were subjected to mud-bath treatment. A cycle of mud bath applications showed beneficial effects on FM patients whose evaluation parameters remained stable after 16 weeks in comparison to baseline [122]. In patients suffering from FM, mud bathing was also found to prevent muscle atrophy and inflammation and improve nutritional condition [123]. Nevertheless, despite positive results, the methodological limitations of available clinical studies, such as the lack of placebo double-blinded trials, preclude definitive conclusions on the effect of body-warming therapies to treat FM [119,124].

A remedy widely used in sports related trauma is the application of cold as a therapeutic agent for pain relief. Cryotherapy refers to the use of low temperatures to decrease the inflammatory reaction, including oedema [125]. Cryotherapy induces several organism physiological reactions like increasing anti-inflammatory cytokines, beta-endorphins, ACTH, white blood cells, catecholamines and cortisol, immunostimulation due to noradrenalin response to cold, the increase in the level of plasma total antioxidant status and the reduction of pain through the alteration of nerve conduction [126]. When compared to control FM subjects, cryotherapy-treated FM patients reported a more pronounced improvement of the quality of life [127]. Whole body cryotherapy was also found to be a useful adjuvant therapy for FM [126].

#### 3.2.5. Hyperbaric Treatment

Hyperbaric oxygen therapy (HBOT) has shown beneficial effects for the prevention and treatment of pain [128], including migraine, cluster headache [129] and FM [130]. HBOT is supposed to induce neuroplasticity that leads to repair of chronically impaired brain functions. HBOT was also found to it improve the quality of life in post-stroke patients and mild traumatic brain injury patients [131]. Therefore, the increased oxygen concentration caused by HBOT is supposed to change the brain metabolism and glial function with a potential effect on reducing the FM-associated brain abnormal activity [132]. HBOT was found to affect the mitochondrial mechanisms resulting in functional brain changes, stimulate nitric oxide production thus alleviating hyperalgesia and promoting the NO-dependent release of endogenous opioids which appear to be involved in the antinociception prompted by HBOT [133]. In a clinical study, a significant difference between the HBOT and control groups was found in the reduction in tender points and VAS scores after the first and fifteenth therapy sessions [130]. These results indicate that HBOT may play an important role in managing FM.

#### 3.2.6. Laser Therapy and Phototherapy

The use of different light wavelengths has been found to be an alternative therapy for FM. It is known that low-level laser therapy is a therapeutic factor, being able not only to target one event in the painful reception, but rather the extend its effectiveness on the whole hierarchy of mechanisms of its origin and regulation [134]. Laser photobiomodulation therapy has been reported to be effective in the treatment of a variety of myofascial musculoskeletal disorders, including FM [135]. The combination of laser therapy and the administration of the drug amitriptyline was found to be effective on clinical symptoms and quality of life in FM; furthermore, gallium-arsenide laser therapy was found to be a safe and effective treatment which can be used as a monotherapy or as a supplementary treatment to other therapeutic procedures in FM [136]. Evidence supported also the use of laser therapy in women suffering FM to improve pain and upper body range of motion, ultimately reducing the impact of FM [137,138]. Finally, a combination of phototherapy and exercise training was evaluated in patients with FM in a randomized controlled trial for chronic pain to offer valuable clinical evidence for objective assessment of the potential benefits and risks of procedures [139].

#### 3.2.7. Exercise and Massage

Exercise therapy seems to be an effective component of treatment, yielding improvement in pain and other symptoms, as well as decreasing the burden of FM on the quality of life [140]. Exercise is generally acceptable by individuals with FM and was found to improve the ability to do daily activities and the quality of life and to decrease tiredness and pain [141]. However, it is important to know the effects and specificities of different types of exercise. For instance, two or more types of exercise may combine strengthening, aerobic or stretching exercise; however, there is no substantial evidence that mixed exercise may improve stiffness [142]. Quality of life may be improved by muscle stretching exercise, especially with regard to physical functioning and pain, whereas depression is reduced by resistance training. A trial including a control group and two intervention groups, both of which receiving exercise programs created specifically for patients with FM, showed that both modalities were effective in an exercise therapy program for FM [143]. A progressive muscle strengthening activity was also found to be a safe and effective mode of exercise for FM patients [144]. Furthermore, strength and flexibility exercises in aerobic exercise rehabilitation for FM patients led to improvements in patients’ shoulder/hip range of motion and handgrip strength [145]. Among women with FM, the association between physical activity and daily function is mediated by the intensity of musculoskeletal pain, rather than depressive symptoms or body mass [146], with a link between clinical and experimental pain relief after the performance of isometric contractions [147].

A randomized controlled trial evaluated the effects of yoga intervention on FM symptoms. Women performing yoga showed a significant improvement on standardized measures of FM symptoms and functioning, including fatigue, mood and pain, and in pain acceptance and other coping strategies [148]. Moreover, the combination with massage therapy program during three months influenced perceived stress index, cortisol concentrations, intensity of pain and quality of life of patients with FM [149].

In terms of societal costs and health care costs, quality of life and physical fitness in females with FM was improved by aquatic training and subsequent detraining [150,151]. Aquatic physical training was effective in promoting increased oxygen uptake at peak cardiopulmonary exercise test in women with FM [152]. A systematic evaluation of the harms and benefits of aquatic exercise training in adults with FM showed that it may be beneficial for improving wellness, symptoms, and fitness in adults with FM [153,154].

A safe and clinically efficacious treatment of pain and other FM symptoms was also achieved by the combination of osteopathic manipulative medicine and pharmacologic treatment with gabapentin [155].

Dancing is a type of aerobic exercise that may be used in FM alternative therapy. Belly dancing was found to be effective in improving functional capacity, pain, quality of life and improving body image of women with FM [156]. More recently, three months treatment of patients with FM with Zumba dancing was found to be effective in improving pain and physical functioning [157].

Finally, Tai chi mind-body treatment was found to improve FM symptoms as much as aerobic exercise and longer duration of Tai chi showed greater improvement. According to a recent report, mind-body approaches may take part of the multidisciplinary management of FM and be considered an alternative therapeutic option [158].

#### 3.2.8. Probiotics and FM Therapy

A tractable strategy for developing novel therapeutics for complex central nervous system disorders could rely on the so called microbiota-gut-brain axis management, because intestinal homeostasis may directly affect brain functioning [159,160]. The pain intensity of patients with FM has been reported to be correlated with the degree of small intestinal bacterial overgrowth, which is often associated with an increased intestinal permeability whose values were significantly increased in the FM patients [161]. Preclinical trials indicate that the microbiota and its metabolome are likely involved in modulating brain processes and behaviors [162]. Therefore, FM patients should show better performance after the treatment with probiotics. In a double-blind, placebo-controlled, randomized design probiotic improved impulsive choice and decision-making in FM patients, but no other effects were observed on cognition, quality of life, self-reported pain, FM impact, depressive or anxiety symptoms [163].

#### 3.2.9. Use of Plant Extracts and Natural Products for FM Treatment

About 40% of drugs used to treat FM originate from natural products [164]; however, there are a few studies that prove the safe and effective use of various plant extracts in FM therapy. Several plant extracts are currently used for their antinociceptive properties and potential to treat FM [165]. 

*Papaver somniferum* is probably the most ancient plant used for its antinociceptive properties [166], with chemical components able to interact with opioid receptors; among these morphine (**15**) which is not only the oldest, but is still the most effective drug for the management of severe pain in clinical practice [167]. The use of opioids for FM treatment has been discussed above.

Another important plant is *Cannabis sativa*. The major active constituent of Cannabis, Δ9-THC (**1**), has been shown to possess antinociceptive properties when assessed in several experimental models [168] (see also the discussion above on cannabinoids). Although there is still scarce evidence to support its role in the treatment of FM, a large consensus indicates that medical cannabis could be an effective alternative for the treatment of FM symptoms [169]. The illicit use of herbal cannabis for FM treatment has been correlated to the inefficacy of current available medications, but is also linked to popular advocacy or familiarity with marijuana from recreational use. Therefore, physicians are requested to examine the global psychosocial well-being, and not focus only on the single outcome measure of pain [52,170]. Although medical cannabis treatment has a significant favorable effect on patients with FM, 30% of patients experience adverse effects [171] and 8% report dependence on cannabis [172]. VAS scores measured in 28 FM patients after 2 hours of cannabis use showed enhancement of relaxation and feeling of well-being, a reduction of pain and stiffness which were accompanied by an increase in somnolence. The mental health component summary score of the Short Form 36 Health Survey was higher in cannabis users than in non-users [49].

Among terpenoids, administration of *trans*-β-caryophyllene (BCP, **26**), a bicyclic sesquiterpene compound existing in the essential oil of many plants like *Copaifera langsdforffii*, *Cananga odorata*, *Humulus lupulus*, *Piper nigrum* and *Syzygium aromaticum*, which provide a high percentage of BCP along with interesting essential oil yields [173], significantly minimized the pain in both acute and chronic pain models [174]. BCP selectively binds to the cannabinoid 2 (CB_2_) receptor and is a functional CB_2_ agonist. Upon binding to the CB_2_ receptor, BCP inhibits adenylate cylcase, leads to intracellular calcium transients and weakly activates the mitogen-activated kinases Erk1/2 and p38 in primary human monocytes [175]. BCP, a safe compound with toxicity at doses higher than 2000 mg/kg body weight [176], was found to reduce the primary and secondary hyperalgesia produced by a chronic muscle pain model (which is considered to be an animal model for FM) [177]. Significant and dose-dependent antinociceptive response was produced by BCP without the presence of gastric damage [178]. Antiallodynic actions of BCP are exerted only through activation of local peripheral CB2 [179]. In neuropathic pain models, BCP reduced spinal neuroinflammation and the oral administration was more effective than the subcutaneously injected synthetic CB2 agonist JWH-133 [180]. Recently, BCP was found to exert an analgesic effect in an FM animal model through activation of the descending inhibitory pain pathway [181]. Thus, BCP may be highly effective in the treatment of long-lasting, debilitating pain states, suggesting the interesting application of BCP in FM therapy.

The analgesic properties of myrrh (*Commiphora myrrha*) have been known since ancient times and depend on the presence of bioactive sesquiterpenes with furanodiene skeletons which are able to interact with the opioid receptors [182,183]. *C. myrrha* extracts exerted a stronger suppression on carrageenan-induced mice paw edema with significant analgesic effects [184] and were effective against chronic inflammatory joint disease such as osteoarthritis [185]. In a preclinical trial, pain alleviation was obtained with *C. myrrha* extracts for many pathologies [186], indicating that extracts from this plant may have the potential to treat FM.

Preclinical studies indicate a potential use of *Hypericum perforatum* (Hypericaceae), popularly known as St. John’s wort, in medical pain management [187] due to its phenolic compounds. Many phenolic compounds (e.g., flavonoids) from medicinal plants are promising candidates for new natural analgesic drugs [188]. Quercetin (**27**) showed analgesic activity and could reduce neuropathic pain by inhibiting mTOR/p70S6K pathway-mediated changes of synaptic morphology and synaptic protein levels in spinal dorsal horn neurons of db/db mice [189], while rutin (**28**) could inhibit the writhing response of mice induced by potassium antimony tartrate and showed to be a promising pharmacological approach to treat pain [190]. The analgesia potency of hyperin (**29**) was approximately 20-fold of morphine, while luteolin (**30**) presented effective analgesic activities for both acute and chronic pain management. Some glycosides of kaempferol (e.g., kaempferol 3-O-sophoroside, **31**) possess significant analgesic activity in the tail clip, tail flick, tail immersion, and acetic acid-induced writhing models, whereas baicalin (**32**) shows analgesic effects in several kinds of pain [191]. Fisetin (**33**), a plant flavonoid polyphenol, has been reported to possess potent antioxidant, antinociceptive and neuroprotective activities. In rats, fisetin acts via modulation of decreased levels of biogenic amines and elevatesoxido-nitrosative stress and ROS to ameliorate allodynia, hyperalgesia, and depression in experimental reserpine-induced FM [192].

In a double-blind parallel-group clinical trial, outpatients with FM were randomized to receive either 15 mg of *Crocus sativus* (saffron) extract or 30 mg duloxetine (**22**). No significant difference was detected for any of the scales neither in terms of score changes from baseline to endpoint between the two treatment arms, indicating that saffron and duloxetine had comparable efficacy in treatment of FM symptoms [193].

It is still unclear the efficacy of natural products extracted from plants in treating FM. However, some clinical data show promising results and more studies with adequate methodological quality are necessary in order to investigate the efficacy and safety of natural products as a support in FM therapy. Figure 7 depicts the chemical formulae of some antinociceptive natural products.

## 4. Conclusions

Diagnosis of FM is based on clinical feature and criteria that still lack either a gold standard or at least supportive laboratory findings. FM diagnostic criteria may include heterogeneous patients also in clinical trials and this may impair evaluation of clinically meaningful treatment effect. 

The review of the literature suggests that a multidisciplinary therapeutic approach, based on the combination of pharmacologic and alternative therapy (including thermal, light, electrostimulatory and body exercise treatments) could improve the quality of life and reduce pain and other symptoms related to FM. However, sometimes the ability of patients to participate to alternative therapies is impeded by the level of pain fatigue, poor sleep, and cognitive dysfunction. These patients may need to be managed with medications before initiating nonpharmacologic therapies.

Although the use of some natural phytochemicals like BCP and phenolic compounds might replace other natural products such as Δ9-THC, because of reduced side effects and higher tolerability, FM self medication practice may be ineffective and in some cases even detrimental. Therefore, providing FM patients with the correct information about their disorders may help monitoring pharmacological and alternative therapies. At the same time maintaining information will help patients to receive the appropriate medications and therapies [194].

## Figures and Tables

**Figure 1 ijms-21-07877-f001:**
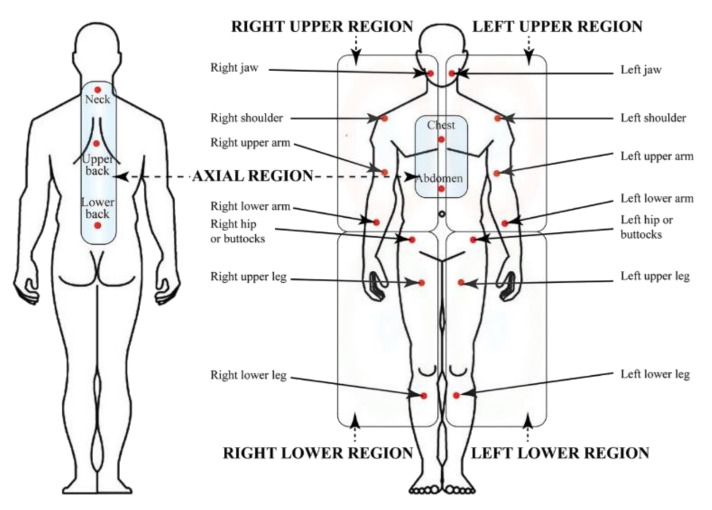
Widespread Pain Index from ACR 1990 criteria for the classification of fibromyalgia and related regions.

**Figure 2 ijms-21-07877-f002:**
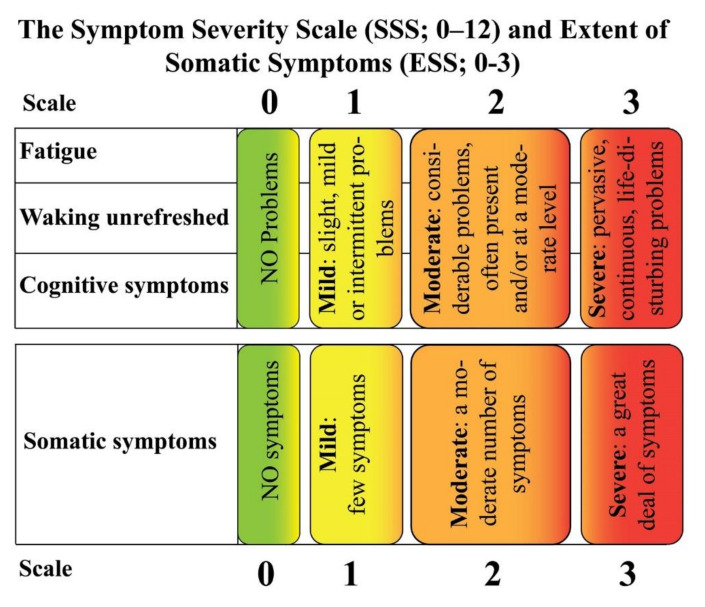
Symptom Severity scale (SSS) and Extent of Somatic Symptoms (ESS).

**Figure 3 ijms-21-07877-f003:**
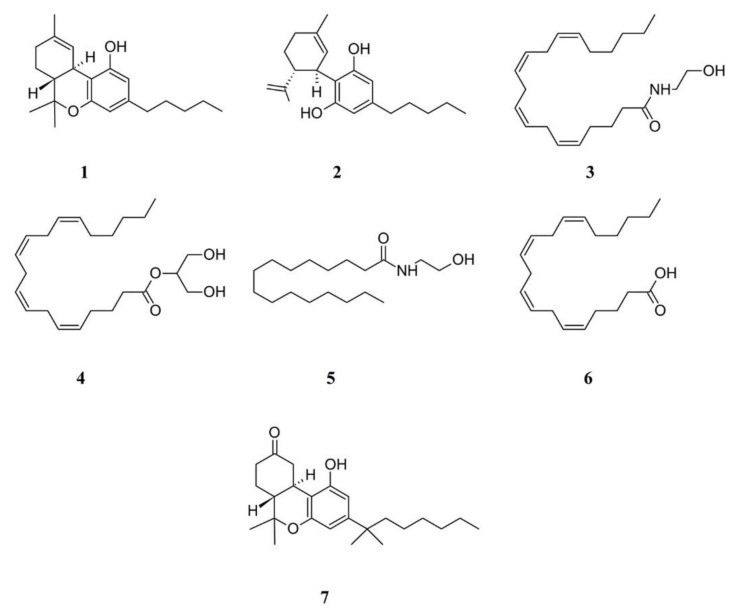
Structure formulae of some cannabinoids and related compounds. Numbers correspond to compound names cited in the text.

**Figure 4 ijms-21-07877-f004:**
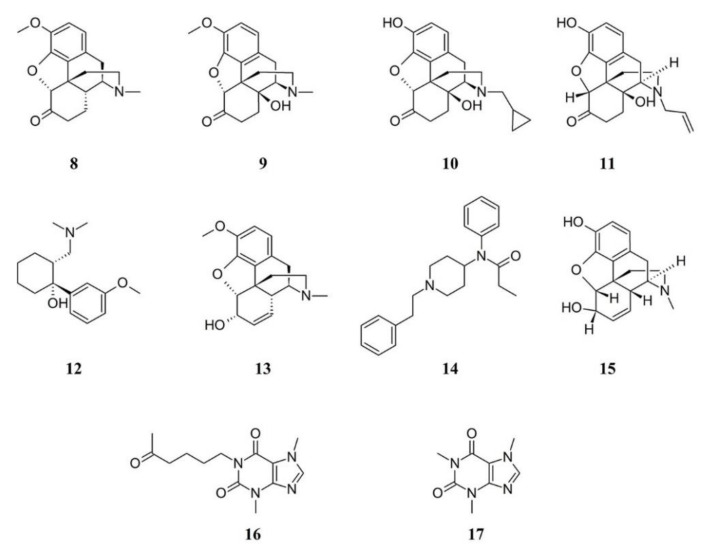
Structure formulae of some opioids and related compounds. Numbers correspond to molecules cited in the text.

**Figure 5 ijms-21-07877-f005:**
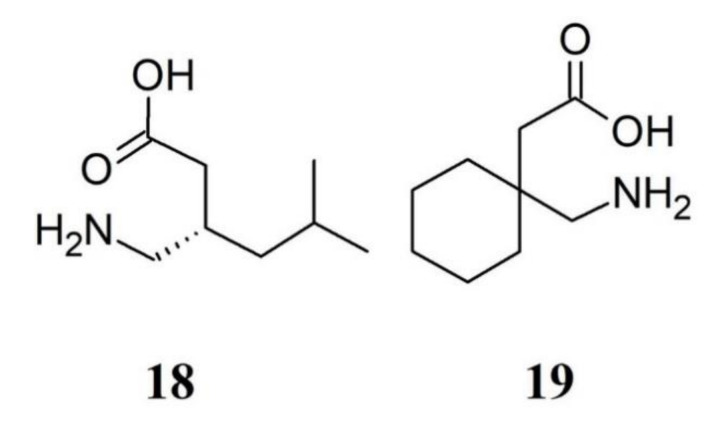
Structure formulae of some gabapentinoids. Numbers correspond to molecules cited in the text.

**Figure 6 ijms-21-07877-f006:**
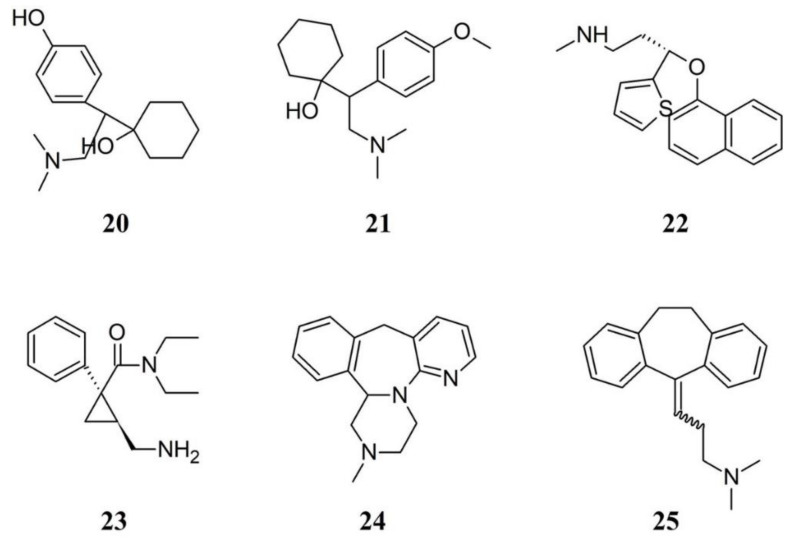
Chemical structure of some serotonin and noradrenaline reuptake inhibitors and a tricyclic antidepressant. Numbers correspond to molecules cited in the text.

**Figure 7 ijms-21-07877-f007:**
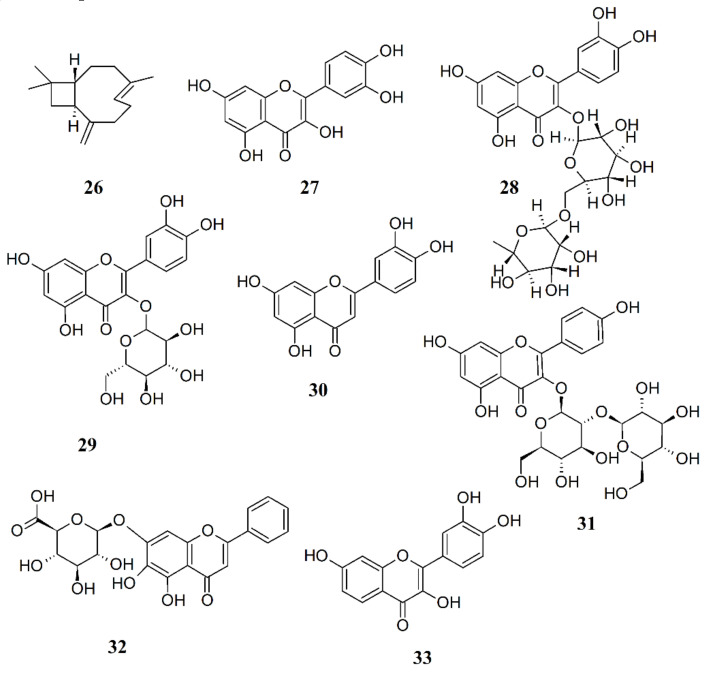
Chemical structure of some natural compounds with antinociceptive activity.

**Table 1 ijms-21-07877-t001:** ACR2010 and modified criteria for the diagnosis of fibromyalgia.

**Widespread pain index (WPI)**			
***Areas***	*specification*		
**Number of areas in which the patient has had pain over the past week**	0–19 points		
Areas to be considered	shoulder girdle, hip (buttock, trochanter), jaw, upper back, lower back, upper arm, upper leg, chest, neck, abdomen, lower arm, and lower leg (all these areas should be considered bilaterally)		
**Symptom Severity Scale (SSS) score**			
*Symptom*	*Level of severity*	*Symptom level*	*Score*
Fatigue Waking unrefreshed Cognitive symptoms (e.g., working memory capacity, recognition memory, verbal knowledge, anxiety, and depression)	For each of these 3 symptoms, indicate the level of severity over the past week using the following scale: 0 = no problem 1 = slight or mild problems, generally mild or intermittent 2 = moderate; considerable problems, often present and/or at a moderate level 3 = severe; pervasive, continuous, life-disturbing problems	Considering somatic symptoms in general, indicate whether the patient has the following: 0 = no symptoms 1 = few symptoms 2 = a moderate number of symptoms 3 = a great deal of symptoms	Final score between 0 and 12
**Criteria**			
*Specification*	*Conditions*		
A patient satisfies diagnostic criteria for fibromyalgia if the following 3 conditions are met	(a)WPI ≥ 7/19 and SS scale score ≥ 5 or WPI 3–6 and SS scale score ≥ 9 (b) symptoms have been present as a similar level for at least 3 months (c) the patient does not have a disorder that would otherwise explain the pain		
**Modified criteria**			
*Specification*	*Conditions*	*Final Score*	
A patient satisfies diagnostic criteria for fibromyalgia if the following 3 conditions are met	(a)WPI (as above) (b) SS scale score (as above, but without extent of somatic symptoms) (c) presence of abdominal pain, depression, headaches (yes = 1, no = 0)	The number of pain sites (WPI), the SS scale score, and the presence of associated symptoms are summed to give a final score between 0 and 31	

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
