# Peer review of "Fibromyalgia: Recent Advances in Diagnosis, Classification, Pharmacotherapy and Alternative Remedies"

_ijms, 2020, doi:10.3390/ijms21217877_

Round 1
Reviewer 1 Report
Dear authors, congratulations for the manuscript.
In my review I have some clarifications that I present below
1.- The new classification of the FM should be specified in the ICD11
2.- Add pathophysiology theories (among others with reference 190)
3.- Follow the same scheme in all evidence treatments
4.- Exercise must be extracted from alternative therapies, it is a non-pharmacological therapy
5.- Remove Papaver somniferum and Cannabis sativa from plants (in any case include in the corresponding drugs)
6.- In my opinion Conclusion must be remake:
In the conclusions there are sentences that have not been discussed or analyzed in the text, include them in the text or withdraw
The proposed treatment is not based on evidence, make it clear that it is an opinion of the author: "A multidisciplinary therapeutic approach, based on the combination of pharmacologic and alternative therapy (including thermal, light, electrostimulatory and body exercise treatments) was found to improve the quality of life and reduce pain and other symptoms related to FM. However, sometimes the ability of patients to participate to alternative therapies is impeded by the level of pain fatigue, poor sleep, and cognitive dysfunction. These patients may need to be managed with medications before initiating nonpharmacologic therapies. Moreover, although the use of phytochemiscals like BCP and phenolic compounds may replace other natural products such as Δ9-THC, because of reduced side effects and higher tolerability, FM self medication practice may be ineffective and in some cases even detrimental."
Author Response
1.- The new classification of the FM should be specified in the ICD11
R: A new sentence discusses the new classification of the ICD-11
2.- Add pathophysiology theories (among others with reference 190)
R: physiopatholy discussion has been extended in the opioid section
3.- Follow the same scheme in all evidence treatments
R: The scheme has been followed as much a s possible
4.- Exercise must be extracted from alternative therapies, it is a non-pharmacological therapy
R: Exercise is included in the subsection 3.2 that includes alternative non-pharmacological therapies
5.- Remove Papaver somniferum and Cannabis sativa from plants (in any case include in the corresponding drugs)
R: These two plants have been inserted among the plant extracts owing to their importance. The discussion of drugs derived from these plants has been treated in the sections of opioids and cannabinoids
6.- In my opinion Conclusion must be remake:
R: conclusions have been revised by removing sentences and concepts not discussed in the main text.
Reviewer 2 Report
The review is well organized and describes both diagnostic and theraupetic FM aspects.
Comments:
1)In the FM diagnosis section, I suggest to add new references concerning molecular and cellular biology finding from recent articles:
a)"Identification of MOR-Positive B Cell as Possible Innovative Biomarker (Mu Lympho-Marker) for Chronic Pain Diagnosis in Patients with Fibromyalgia and Osteoarthritis Diseases. Int J Mol Sci. 2020 Feb 22;21(4):1499."
b)Metabolic fingerprinting for diagnosis of fibromyalgia and other rheumatologic
disorders. J. Biol. Chem. 2019, 294, 2555–2568.
where the authors talk about new diagnostic strategies for FM diagnosis.
2)Table1 appears graphically confusing, I suggest to better re-organize it in the page.
3)Line 46-47: "Despite advances in pathologic processes understanding, undiagnosed FM remains in as many as 75% of people with the condition [8]."
I suggest to re-phrase, it is not clear and well written.
Author Response
1)In the FM diagnosis section, I suggest to add new references concerning molecular and cellular biology finding from recent articles: a)"Identification of MOR-Positive B Cell as Possible Innovative Biomarker (Mu Lympho-Marker) for Chronic Pain Diagnosis in Patients with Fibromyalgia and Osteoarthritis Diseases. Int J Mol Sci. 2020 Feb 22;21(4):1499." b)Metabolic fingerprinting for diagnosis of fibromyalgia and other rheumatologic disorders. J. Biol. Chem. 2019, 294, 2555–2568. where the authors talk about new diagnostic strategies for FM diagnosis.
R: I thank the reviewer for suggesting these two important articles that have been inserted and discussed at the end of the Diagnosis section
2)Table1 appears graphically confusing, I suggest to better re-organize it in the page.
R: I have framed the table for a better reading and I am sure that the production operators will make it framed in a perfect way
3)Line 46-47: "Despite advances in pathologic processes understanding, undiagnosed FM remains in as many as 75% of people with the condition [8]." I suggest to re-phrase, it is not clear and well written.
R: the sentence has been rephrased as requested